# NIR-II bioluminescence for in vivo high contrast imaging and in situ ATP-mediated metastases tracing

Lingfei Lu [1], Benhao Li[1], Suwan Ding [1], Yong Fan [1✉], Shangfeng Wang[1], Caixia Sun[1], Mengyao Zhao[1], Chun-Xia Zhao [2] & Fan Zhang [1✉]

Bioluminescence imaging has been widely used in life sciences and biomedical applications. However, conventional bioluminescence imaging usually operates in the visible region, which hampers the high-performance in vivo optical imaging due to the strong tissue absorption and scattering. To address this challenge, here we present bioluminescence probes (BPs) with emission in the second near infrared (NIR-II) region at 1029 nm by employing bioluminescence resonance energy transfer (BRET) and two-step fluorescence resonance energy transfer (FRET) with a specially designed cyanine dye FD-1029. The biocompatible NIR-II-BPs are successfully applied to vessels and lymphatics imaging in mice, which gives ~5 times higher signal-to-noise ratios and ~1.5 times higher spatial resolution than those obtained by NIR-II fluorescence imaging and conventional bioluminescence imaging. Their capability of multiplexed imaging is also well displayed. Taking advantage of the ATP-responding character, the NIR-II-BPs are able to recognize tumor metastasis with a high tumor-to-normal tissue ratio at 83.4.

[1] Department of Chemistry, State Key Laboratory of Molecular Engineering of Polymers and iChem, Shanghai Key Laboratory of Molecular Catalysis and Innovative Materials, Fudan University, 200433 Shanghai, P.R. China. [2] Australian Institute for Bioengineering and Nanotechnology, The University of Queensland, St. Lucia, QLD 4072, Australia. ✉email: fan_yong@fudan.edu.cn; zhang_fan@fudan.edu.cn

Optical imaging with non-invasion, real-time, fast feedback and high sensitivity has played a crucial role concerning in vivo bioinformatics visualization[1–7]. One of the dominant limitations against acquiring high-performance imaging with high signal-to-noise ratios (SNRs) is the autofluorescence background arising from endogenous fluorophores in complex biological organs and tissues, such as melanin, elastin, collagen, keratin, porphyrins and flavins upon excitation by external radiation[8,9]. Therefore, optical imaging strategies such as bioluminescence imaging, chemiluminescence imaging and afterglow imaging have attracted great interest, as they eliminate the demand for simultaneous light excitation. So far, bioluminescence imaging has been widely used for tracking cells[10], monitoring gene expression[11,12], sensing small bioactive molecules[13], and tumor imaging[14]. However, the short emission wavelengths locating in the visible range (VIS, 400–700 nm) of conventional bioluminescence imaging bring about the intrinsic limitation of strong tissue absorption and scattering for in vivo imaging[15], which restricts the detection of signals from deep tissues with high sensitivity and high spatio-temporal resolution. Through bioluminescence resonance energy transfer (BRET)[16], the self-illuminating emission wavelengths in VIS reigon have successfully been extended to the first near-infrared (NIR-I, 700–900 nm) window with low tissue absorption for improved tumor and lymph node imaging as well as molecular imaging of enzyme activities[17–19], but the scattering effect remains an obstacle, blurring the image at larger penetration depth in tissue[20]. Compared with imaging in VIS and NIR-I window, recent works have revealed that optical imaging in the second near infrared (NIR-II, 1000–1700 nm) region can achieve higher spatial resolution in deep tissues (above 1 cm) because the above adverse factors are suppressed[21–37]. One recent clinical study also highlights the promising clinical potential of intraoperative NIR-II fluorescence imaging and NIR-II image-guided surgery[38]. However, the necessity of external illuminations for excitation in in vivo NIR-II fluorescence imaging inevitably leads to other unfavorable outcomes, such as the autofluorescence background, potential light-induced overheating effect and inhomogeneous illumination in wide-field imaging[39–41].

In this work, we report the development of NIR-II bioluminescence probes (NIR-II-BPs) based on a specially designed cyanine dye FD-1029 to overcome the above challenges by integrating a BRET process with a two-step fluorescence resonance energy transfer (FRET) process (Fig. 1). Compared with NIR-II fluorescence imaging on vascular and lymphatic systems, strikingly high SNRs and improvement in spatial resolution are obtained by using NIR-II bioluminescence imaging. The flexibility of the energy transfer strategy also allows the emission wavelength of the bioluminescence probes to be tuned, which enables high-performance multicolor imaging on superficial vessels and tumor tissues simultaneously in a living mouse. In addition, owing to the Adenosine Triphosphate (ATP)-dependent manner of NIR-II-BPs[42], intracellular ATP-mediated imaging in solid and metastatic tumors is performed with high tumor-to-normal tissue (T/N) ratio, demonstrating that the NIR-II-BPs may hold a great promise in metastases tracing.

## Results

**Preparation and characterization of NIR-II-BPs.** The synthesis procedure of NIR-II-BPs is illustrated in Fig. 1. We adopted a BRET-FRET-FRET energy transfer relay process, in which firefly luciferase ($\lambda_{em} = 560$ nm) was chosen as the self-luminescent source. To achieve emission in the state-of-the-art NIR-II region, a specially designed hydrophilic cyanine dye FD-1029 with a heptamethine structure was synthesized (Fig. 2a and Supplementary Fig. 2). The maximum absorption and emission peaks of FD-1029 were at 977 and 1029 nm, respectively (Fig. 2b, c). When encapsulated in amphiphilic polymers DSPE-PEG2000 as micelles, the quantum yield of FD-1029 was 0.029% (Supplementary Table 1). By increasing the dye concentration loaded in micelles, we found that the absorption and emission peaks of FD-1029 stayed constant in a large concentration range (1–10 μM), which was mainly attributed to the large dihedral angle (~60°) between ipsilateral indole groups in FD-1029 (Supplementary Fig. 3) and its good hydrophilicity. However, aggregation happens when higher concentration of FD-1029 is loaded (>10 μM) (Fig. 2b, c). Therefore, the optimal concentration of the dyes is critical for modulating the superior property of the final NIR-II-BPs. In order to bridge the energy transfer gap between luciferase and FD-1029, two commercial organic dyes (Cy 5 and Cy 7.5) were introduced due to the sufficient overlap in emission and

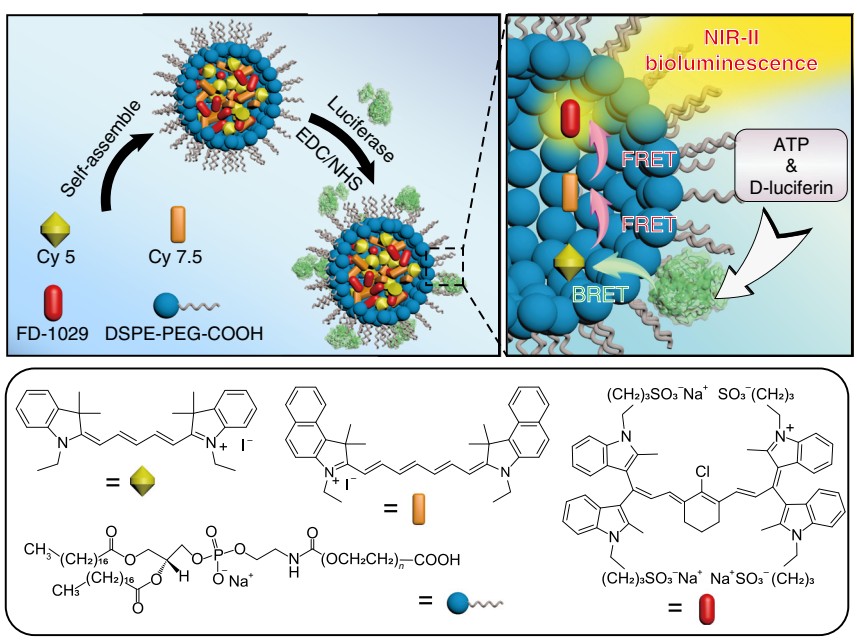

**Fig. 1 Concept of NIR-II-BPs with NIR-II bioluminescence.** Schematic procedures for synthesizing NIR-II-BPs and mechanism of NIR-II bioluminescence.

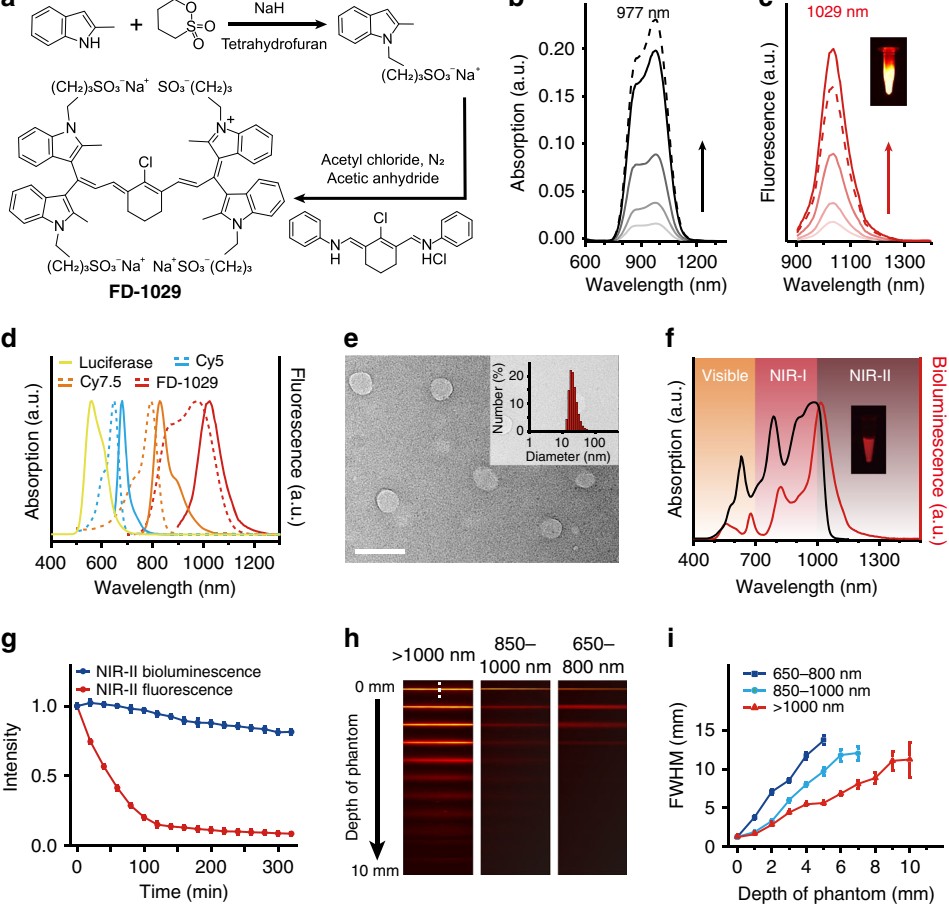

**Fig. 2 Synthesis and characterization of NIR-II-BPs. a** Synthetic route of FD-1029. **b** Absorbance spectra of FD-1029 in DSPE-PEG2000 micelles with the concentration of 1, 2, 5, 10 μM (from bottom to top). The dashed line refers to 12 μM FD-1029 in micelles. **c** Fluorescent emission spectra of FD-1029 in DSPE-PEG2000 micelles with the concentration of 1, 2, 5, 10 μM (from bottom to top). Inset: NIR-II fluorescence image of FD-1029 in DSPE-PEG2000 micelles (10 μM) (1000 nm long-pass filter) under excitation of 980 nm. **d** UV/VIS/NIR absorption and emission spectra of Cy5, Cy7.5, FD-1029, and bioluminescence of luciferase. Dashed and solid lines refer to absorption and emission spectra, respectively. **e** Representative TEM image and size distribution of NIR-II-BPs in PBS buffer. Scale bar, 50 nm. Repeated for three times in independent experiments. **f** Bioluminescence emission spectrum and absorption spectrum of NIR-II-BPs in PBS buffer. **g** Photostability of NIR-II-BPs in mouse serum. Bars represent mean ± s.d. derived from $n = 3$ independent detections. **h** Bioluminescence images of capillaries filled with NIR-II-BPs immersed with 1% intralipid at varied depth in different detection windows. **i** The FWHM of Gaussian fitted intensity data in **g** versus different intralipid depth. Bars represent mean ± s.d. derived from $n = 3$ independent experiments. Source data underlying **g** and **i** are provided as a Source Data file.

absorption bands between the four emitters (Fig. 2d). Through the self-assembly of amphiphilic polymers DSPE-PEG2000-COOH, monodispersed micelles loaded with Cy5, Cy7.5, and FD-1029 were generated with improved biocompatibility and water solubility. After that, luciferase was conjugated to the micelles to produce the final NIR-II-BPs with a diameter of 25.0 ± 6.3 nm (Fig. 2e and Supplementary Fig. 4). The average number of luciferases on a single micelle was estimated to be ~5 (Supplementary Fig. 5 and Supplementary Table 2).

In the as-prepared NIR-II-BPs, a broad UV/VIS/NIR absorption with a maximum at 977 nm was observed (Fig. 2f). Upon the addition of D-luciferin, three weak emissions at 560, 678, 822 nm as well as one strong NIR-II peak at 1029 nm could be clearly identified (Fig. 2f). These emission bands were ascribed to luciferase, Cy5, Cy7.5, and FD-1029, respectively, suggesting the successful BRET-FRET-FRET from luciferase to FD-1029. It should be noted that the Cy5 and Cy 7.5 were indispensable in bridging the cascade energy transfer from luciferase to the final emission of FD-1029. (Supplementary Fig. 4E). To minimize self-quenching effect and realize the highest energy transfer efficiency, the optimal ratio among the three dyes was carefully modulated

and found to be 1:2:10 (see Supplementary Information for detailed discussion, Supplementary Fig. 6A–D). Under the circumstance, FRET efficiencies between Cy5&Cy7.5 and Cy7.5&FD-1029 were determined to be 64.1% and 90.8%. The Förster distances ($R_0$) of these two steps were also calculated to be 1.3 nm and 6.4 nm, respectively (Supplementary Fig. 6A, B), which guarantee the efficient FRET between these dyes and bridge the energy to the NIR-II region. In addition, the overall BRET ratio of NIR-II-BPs was calculated to be 4.20 through dividing the intensity of acceptor emission (625–1400 nm) by that of the donor emission (500–625 nm) (Fig. 2f)[16].

Thanks to the lipid shells, the NIR-II-BPs were stable for a week in both PBS buffer and mouse serum without obvious aggregation (Supplementary Fig. 7). Once D-luciferin was added, the NIR-II bioluminescence signal was fairly stable, with over 80% of the maximum intensity remained after 320 min (Fig. 2g) and still 54% left over 24 h at 37 °C (Supplementary Fig. 8A). Besides, NIR-II-BPs also exhibited good bioluminescence stability after stored in mouse serum for one week at various temperatures (Supplementary Fig. 8B–E). Since NIR-II-BPs were applicable for NIR-II fluorescence as well, photostability was also tested on this

mode for comparison under continuous irradiation of 808 nm light, in which case a sharp decline to 19.4% in the first 90 min was observed and only 8.8% of the initial NIR-II fluorescence intensity left after 320 min (Fig. 2g).

We next studied the penetration depths and spatial-resolutions of bioluminescence in three detection windows (650–800, 850–1000, and >1000 nm, separated by different filters) using 1% Intralipid as a mimic of tissue. Bioluminescence imaging of a capillary tube filled with the NIR-II-BPs solution was acquired by using a home-made VIS and NIR-II imaging system (Supplementary Fig. 9A, B). As shown in Fig. 2h, the visibility of the capillaries deteriorated as the tissue depth increased from 0 to 10 mm for all the three detection windows. Shallower penetration depths (<5 mm) along with severe scattering of photons were observed in two short wavelength windows (Fig. 2i and Supplementary Fig. 9C). In contrast, the NIR-II bioluminescence imaging taken beyond 1000 nm showed sharp edge of capillary with a large penetration depth (10 mm) (Fig. 2h and

Supplementary Fig. 9C, D). These facts suggested that the NIR-II-BPs with stable NIR-II bioluminescence signals were suitable for in vivo deep tissue imaging applications.

**Influences of external excitation on in vivo imaging.** In conventional fluorescence imaging process, external excitation light (from UV to near infrared light) is necessity and will inevitably cause autofluorescence background in organs and tissues. We first evaluated the SNRs of a capillary tube filled with NIR-II-BPs solution, which was covered with chicken breast tissue at different thicknesses, via both NIR-II bioluminescence imaging and NIR-II fluorescence imaging (Fig. 3a). While tissue autofluorescence signal can be clearly detected in NIR-II fluorescence imaging (Fig. 3b, top), this background noise is basically eliminated in the absence of external light with NIR-II bioluminescence imaging (Fig. 3b, bottom). This leads to enhanced SNRs (1.5–3 times) throughout all the different thicknesses of tissues (up to 10 mm) during NIR-II

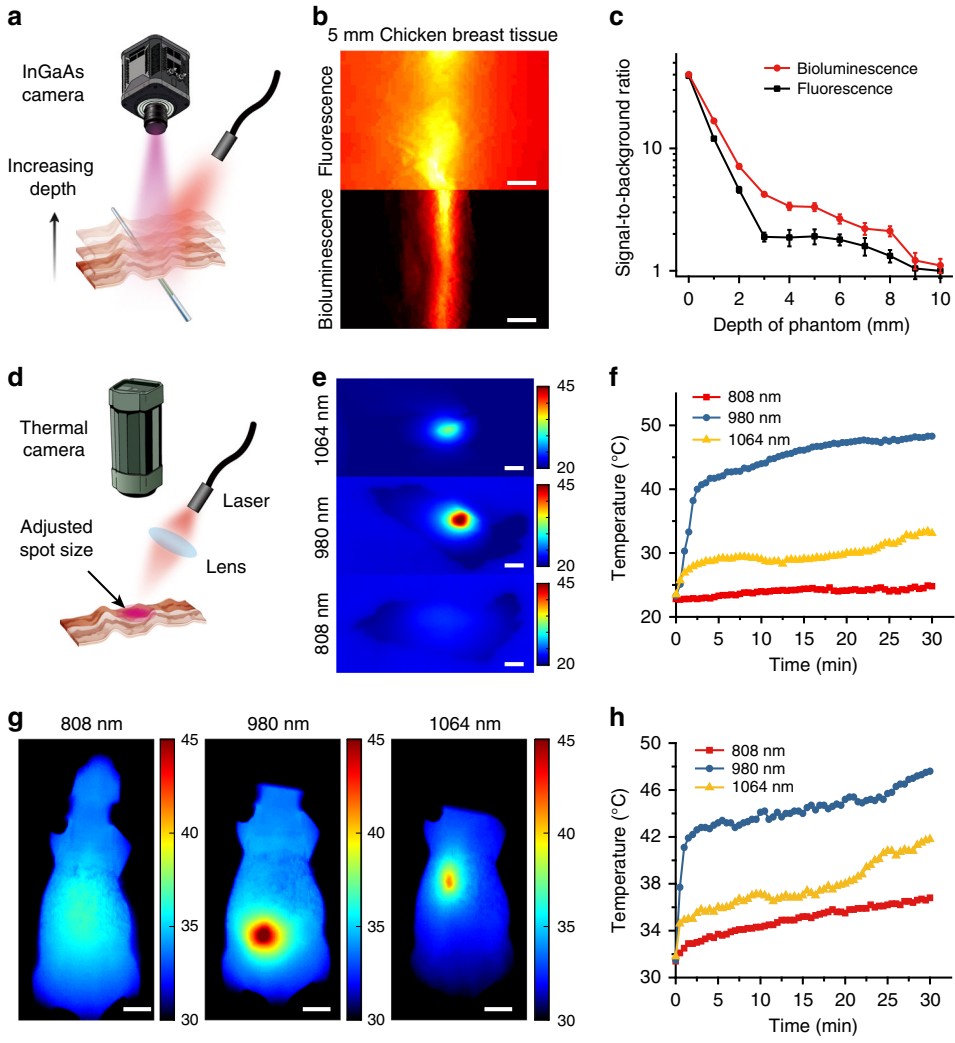

**Fig. 3 Differences in signal-to-noise ratios and photothermal effects between NIR-II bioluminescence and fluorescence imaging. a** Schematic illustration of optical imaging. **b** NIR-II fluorescence image (top) and NIR-II bioluminescence image (bottom) under 5 mm chicken breast tissue. Scale bar, 2 mm. **c** Signal-to-noise ratios of capillaries filled with NIR-II-BPs with different thicknesses of chicken breast tissue for NIR-II fluorescence imaging and NIR-II bioluminescence imaging. Bars represent mean ± s.d. derived from $n = 3$ independent experiments. **d** Schematic illustration for thermal effect study on chicken breast tissues. **e** Photothermal imaging of chicken breast tissue with continuous irradiation under 808 nm (329 mW cm$^{-2}$), 980 nm (726 mW cm$^{-2}$) and 1064 nm (1070 mW cm$^{-2}$) for 30 min. Scale bar, 5 mm. **f** Temperature variation curves of chicken breast tissue recorded with irradiation of different excitation wavelengths. **g** Skin temperatures of nude mouse irradiated under 808 nm (329 mW cm$^{-2}$), 980 nm (726 mW cm$^{-2}$) and 1064 nm (1070 mW cm$^{-2}$) after 30 min. Scale bar, 1 cm. **h** Temperature variation curves of nude mouse skin recorded with irradiation of different excitation wavelengths. Source data underlying **f**, **h** are provided as a Source Data file.

bioluminescence imaging as compared with that in NIR-II fluorescence imaging (Fig. 3c), showing an improvement in imaging clarity.

In addition, overheating effect from the excitation light may also be a potential adverse factor for in vivo imaging. We chose three light sources (808 nm, 980 nm and 1064 nm), which are all able to excite NIR-II-BPs, to study the photothermal effects on chicken breast tissues under their maximum permissible exposure limits[43] (Fig. 3d). Resulting from a strong water absorption peak at 980 nm, the temperature of the tissue quickly increased from 22.5 °C to 42.5 °C within 5 min and continued to increase over time if 980 nm light was applied (Fig. 3e, f). In comparison, the temperature of the tissue showed ~3 °C and ~7 °C increase when exposed to 808 nm and 1064 nm laser, respectively (Fig. 3e, f) due to the reduced absorption coefficient of tissues to the light. In the in vivo photothermal study (Fig. 3g, h), the skin temperature of nude mouse increased to 48ºC within 30 min under the irradiation of 980 nm laser, which could be destructive and fatal to the living body[44]. Similarly, laser irradiation of 808 nm and 1064 nm also caused photothermal effect on the living body within the same period, causing the maximum temperature of the mouse skin to increase by ~ 5 °C (Fig. 3g, h). Nevertheless, the photothermal effect may show a disadvantage to the photostability of NIR-II-BPs. For example, compared with the signal from NIR-II bioluminescence imaging, only half of the initial intensity remained after irradiation of 808 nm (30 mW cm⁻²) light for 30 min in NIR-II fluorescence imaging (Fig. 2g). These results uncovered the latent harm of unexpected overheating effect caused by irradiation, which should be reconsidered and assessed.

Moreover, intensity distribution of the excitation light may also bring distorted signals, especially in wide-field fluorescence imaging. A normal laser beam is usually subjected to a Gaussian point spread function (PSF) and the intensity distribution within the illumination site was expected to decrease from center to periphery with the maximum localized at the center (Fig. 4d). Taking the 808 nm light as an example, the rather low-quality factor (QF < 40) (Fig. 4e) indicates a conspicuous intensity

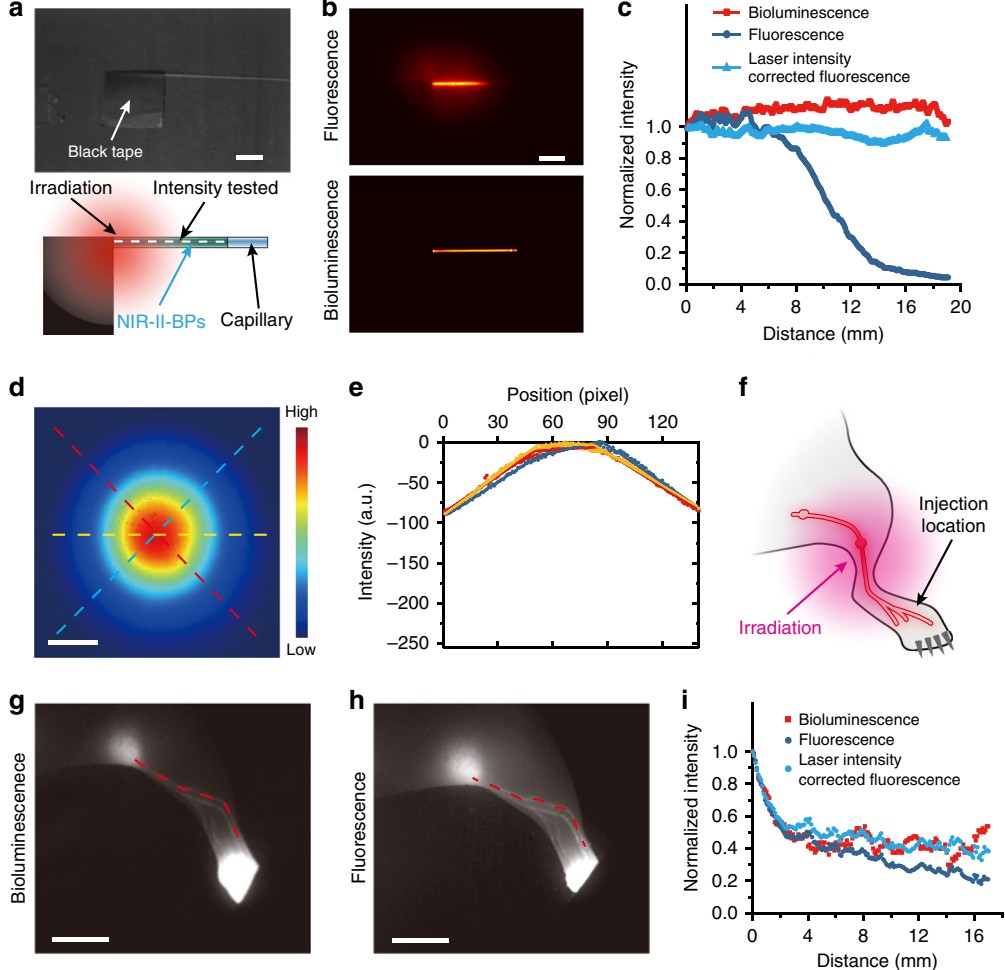

**Fig. 4 Influence of the external excitation light on the signal uniformity of NIR-II bioluminescence and fluorescence imaging. a** Photo (top) and schematic illustration (bottom) of inhomogeneous illumination. Scale bar, 5 mm. **b** Optical images of capillaries filled with NIR-II-bps in NIR-II fluorescence imaging with inhomogeneous illumination (top) and NIR-II bioluminescence imaging (bottom). Scale bar, 5 mm. **c** Intensity profiles along the capillary in g in NIR-II fluorescence imaging and NIR-II bioluminescence imaging. **d** A typical Gaussian point spread function facula. Scale bar, 5 mm. **e** Intensity profiles along the dashed lines with the same color in a Gaussian point spread function in **d**. The positions in the calculations for the dashed lines were from upper left to lower right (red), upper right to lower left (blue), and left to right (yellow). Each scan has been approximated by three optimally determined line segments whose slopes are used in the quality factor (QF) calculation. **f** The illustration of in vivo study on inhomogeneous illumination. The NIR-II bioluminescence (**g**) and NIR-II fluorescence (**h**) imaging on lymphatics in mice after intravenous injection of NIR-II-BPs. Scale bar, 5 mm. **i** Intensity profiles along the red dashed lines in **g**, **h** as well as the corrected intensity profiles considering the intensity distribution of the excitation light. Source data underlying **c**, **e**, **i** are provided as a Source Data file.

variation of the excitation light[45], which would inevitably cause inhomogeneous signals from the fluorophores in deep tissue. We analyzed the emission intensity distribution of a capillary filled with NIR-II-BPs solutions in both NIR-II bioluminescence imaging and NIR-II fluorescence imaging as schematic illustration in Fig. 4a. In NIR-II bioluminescence imaging, the intensity along the capillary tube showed a uniform pattern (Fig. 4b, bottom). However, due to the inhomogeneous illumination of the excitation light, an incongruous intensity distribution along the capillary tube was observed as expected (Fig. 4b, top) in NIR-II fluorescence imaging. Even near the far end of the capillary tube, the NIR-II fluorescence signal was attenuated to almost zero (Fig. 4c). We also injected NIR-II-BPs into the mouse's hind paw, and performed a comparison of NIR-II bioluminescence imaging and NIR-II fluorescence imaging of lymphatic vessels to verify the effect of inhomogeneity of the excitation light source in in vivo imaging (Fig. 4d). From the result of NIR-II fluorescence imaging, it could be seen that the signal intensity decreases significantly as it moved away from the center of the excitation light (Fig. 4e) and the relative intensity at the far end of the irradiation position was only half that of NIR-II bioluminescence imaging, which meant that signal intensity distortion may occur in in in vivo imaging modalities that require external excitation light sources (Fig. 4f). Although calibration with laser power distribution can improve NIR-II fluorescence results (Fig. 4c, i), this may not be easily realized in experiment where the complexity of organs and tissues should be considered. All of the above adverse effects from

external light may hinder the high-resolution and long-time in vivo imaging with deep penetration, but NIR-II bioluminescence without external light can be the better alternative.

**Blood vessels and lymphatics imaging using NIR-II-BPs.** Encouraged by the above results in NIR-II bioluminescence imaging with high penetration depth, high resolution, and high SNR, we applied the NIR-II-BPs to in vivo NIR-II bioluminescence imaging. First, cytotoxicity study of the NIR-II-BPs were evaluated in human ovarian carcinoma cell line CaOV-3 (Supplementary Fig. 10). When incubated with NIR-II-BPs at the concentration of $200 \mu g\ mL^{-1}$ for 24 h, more than 90% of the cells still survived, demonstrating good biocompatibility of the NIR-II-BPs. Next, we performed real-time NIR-II bioluminescence imaging of the mice abdominal vascular and lymphatic ($n = 3$) after a mixture of NIR-II-BPs, ATP, and D-luciferin was applied via tail vein injection immediately. At ~5 min post-injection, strong fluorescence signals from the abdominal vessels and liver as well as tissue autofluorescence background were clearly detected in NIR-II fluorescence imaging (Fig. 5a, right). However, these autofluorescence background noises were greatly reduced in NIR-II bioluminescence imaging (Fig. 5a, left), leading to sharper abdominal vessels with full width at half maxima (FWHM) 1.3 times (imaging depth ~ 1 mm) smaller than that of NIR-II fluorescence imaging (Fig. 5d). Meanwhile, the SNR of the blood vessels also enhanced by almost 5 times through NIR-II

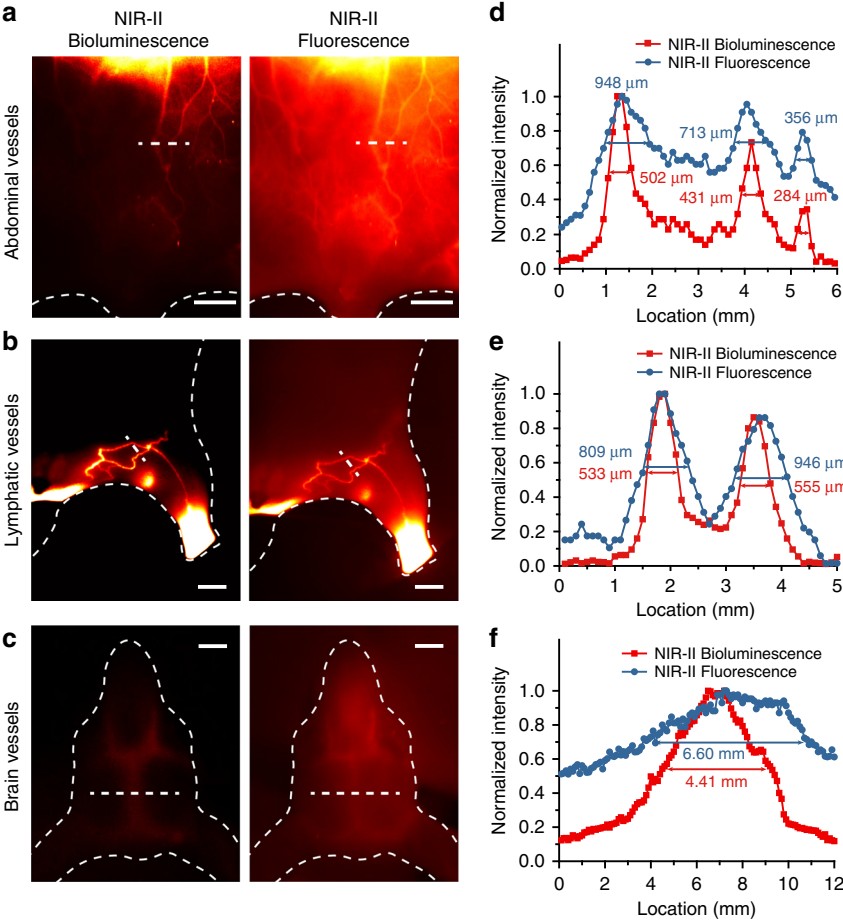

**Fig. 5 Comparing blood vessels and lymphatics imaging between NIR-II bioluminescence and fluorescence signals. a–c** NIR-II bioluminescence imaging (left) and NIR-II fluorescence imaging (right) on abdominal vessels (**a**), lymphatics (**b**) and brain vessels (**c**) in mice after intravenous injection of NIR-II-BPs. Scale bar, 5 mm. **d–f** Intensity profiles of abdominal vessels, lymphatics and brain vessels along the white line in **a–c**. Source data underlying **d–f** are provided as a Source Data file.

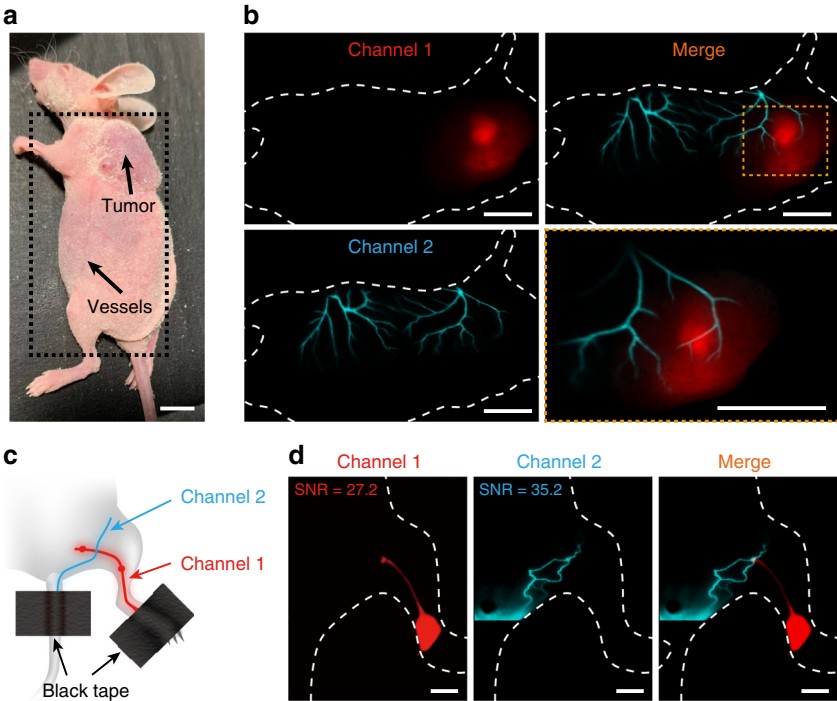

**Fig. 6 In vivo multiplexed imaging using NIR-II-BPs. a** Photograph depicting CT-26 tumor mouse for dual-channel bioluminescence imaging. Scale bar, 10 mm. **b** Dual-channel bioluminescence imaging of tumor tissue (upper left) and vessels (lower left) with NIR-I-BPs and NIR-II-BPs. The zoomed-in high-magnification dual-channel image (lower right) displays fine structure between CT-26 tumor and vessels. Scale bar, 10 mm. **c** Schematic illustration of dual-channel bioluminescence imaging of lymph system. **d** Dual-channel bioluminescence imaging of lymph system with NIR-I-BPs and NIR-II-BPs. Scale bar, 5 mm. (Red channel 1: NIR-I-BPs, Cyan channel 2: NIR-II-BPs).

bioluminescence imaging (20.7) compared with that of NIR-II fluorescence imaging (3.9). Similar results were also achieved in the much sharper lymphatics imaging (FWHM decreased by 1.5 times) with high SNR (increased by 5 times) (Fig. 5b, e) and brain vessels imaging (FWHM decreased 33 % and SNR increased by 3.5 times) (Fig. 5c, f).

**In vivo multiplexed imaging using NIR-II-BPs.** Due to the flexibility of the energy transfer strategy, bioluminescence probes with different emission wavelengths can be developed for high-resolution multicolor in vivo imaging. As a proof of concept, we performed in vivo two-color bioluminescence imaging, motivated by the potential of probing and differentiating multiple components or molecular targets in deep tissues. Similar to NIR-II-BPs, we synthesized a probe without doping FD-1029 (abbreviated as NIR-I-BPs as there was only BRET-FRET process), which gave the peak emission wavelength of Cy7.5 at 822 nm (Supplementary Fig. 6E). By combining NIR-I-BPs and NIR-II-BPs, dual-channel imaging differentiate CT-26 tumor tissues and superficial blood vessels was carried out by hypodermic injection and intravenous injection, respectively (Fig. 6a). The tumor tissues were first highlighted with bioluminescence of NIR-I-BPs between 800 and 900 nm (channel 1) by hypodermic injection and then NIR-II-BPs were intravenous injected into the blood circulation to image the superficial blood vessels with bioluminescence beyond 1000 nm (channel 2) (Fig. 6b, left). After overlaying these two colors, we found the the relative location of CT-26 tumor and vessels with fine structures (Fig. 6b, right).

Moreover, this dual-channel bioluminescence imaging strategy could also be utilized in lymphatic system mapping. By using NIR-II-BPs and NIR-I-BPs, two-channel lymphatic system mapping was achieved (Fig. 6c): the bioluminescence of NIR-I-BPs (channel 1) highlighted the lymphatic from hind paw to popliteal lymph node and bioluminescence of NIR-II-BPs

(channel 2) highlighted the lymphatic from tail base to inguinal lymph node (Fig. 6d). Additionally, both channels showed high SNRs above 27 (Fig. 6d channel 1 = 27.2, channel 2 = 35.2). The merged channel reflected the independence of two channels and unfolded a clear picture of the lower extremity lymphatic system.

**ATP-mediated tumor imaging and metastases tracing.** Due to the good response and selectivity of the NIR-II bioluminescence to ATP (Supplementary Fig. 9), ATP-mediated tumor cell tracing was carried out with the NIR-II-BPs concidering the higher concentrations of ATP in tumor tissues than that in normal tissues[46–48]. We chose human ovarian carcinoma cell CAOV-3 and human embryonic kidney cell HEK293T as the tumor and normal tissue models, respectively. After incubating with both cell lines, NIR-II-BPs could be successfully endocytosed in both cell lines after 3 h (Supplementary Fig. 12). Unsurprisingly, the NIR-II bioluminescence signal exhibited in CAOV-3 cells was three times higher than that of HEK293T and was consistent with that obtained from commercial ATP assay using luciferase (Supplementary Fig. 13). In addition, the metabolic levels in different organs were also evaluated using NIR-II-BPs, which revealed stronger NIR-II bioluminescence signals in the exuberant metabolic organs, such as the brain, heart, liver, and the tumor, confirming the capability of the NIR-II-BPs in studying ATP-related physiological activity. (Supplementary Fig. 14).

Moreover, in situ ATP-mediated tumor tracing was carried out with NIR-II-BPs as well. Nude mice (n = 3) bearing xenografted human ovarian adenocarcinoma tumors were adopted as the model[49]. NIR-II-BPs were first injected intravenously into the nude mice via tail vein. To sufficiently accumulate NIR-II-BPs in the tumor site, a second injection of D-luciferin was performed 24 h later (Supplementary Figs. 15 and 16). After 10 min post-injection of D-luciferin, NIR-II bioluminescence imaging was immediately performed and strong NIR-II bioluminescence

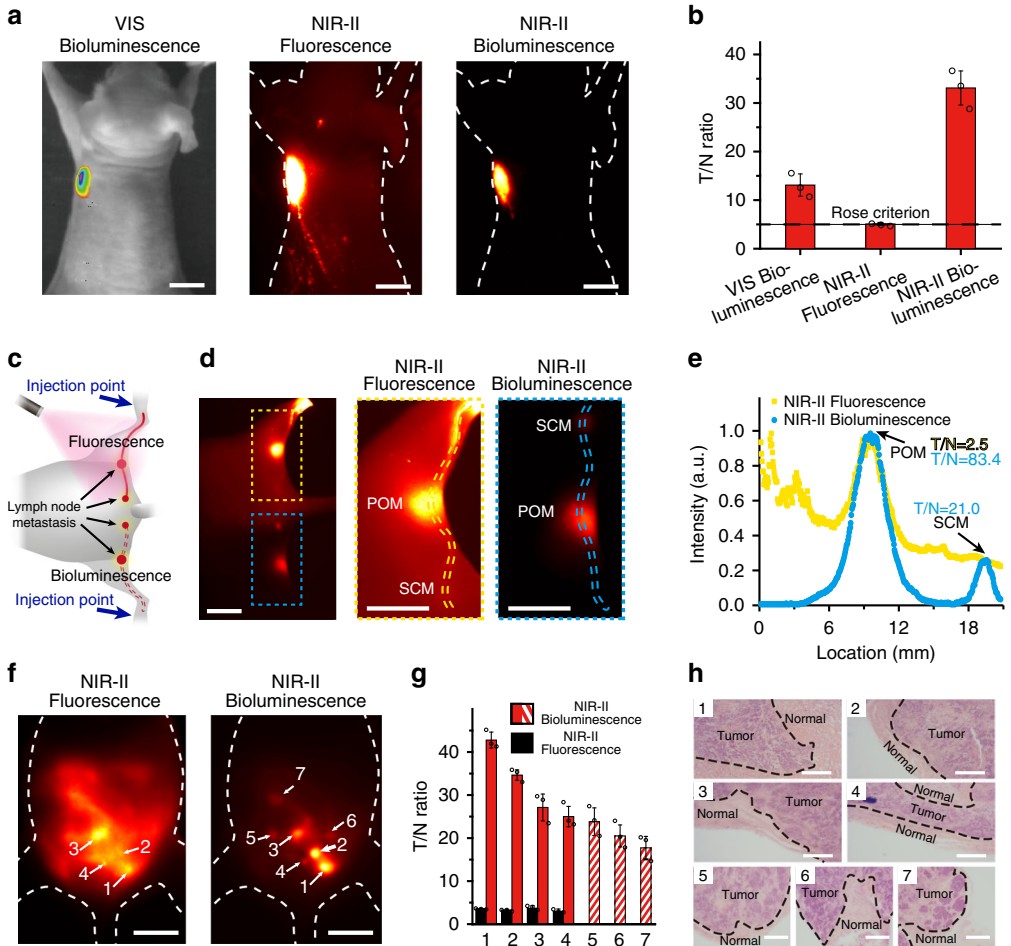

**Fig. 7 ATP-mediated NIR-II-BPs for tumor and metastases imaging. a** VIS bioluminescence imaging, NIR-II fluorescence imaging and NIR-II bioluminescence imaging of the same tumor in a nude mouse. Scale bar, 1 cm. **b** T/N ratios of different optical imaging methods in **a**, black dashed line indicates the Rose criterion. Bars represent mean ± s.d. derived from *n* = 3 independent measurements. **c** Schematic illustration of executing fluorescence imaging and bioluminescence imaging simultaneously on one nude mouse carrying lymph node metastases. Blue arrows indicate the subcutaneous injection location. **d** Fluorescence imaging (top) and bioluminescence imaging (bottom) of lymph node metastasis and the corresponding high-magnification imaging (right). Scale bar, 5 mm. **e** Intensity profiles along the lymphatic vessels in **d**. **f** NIR-II fluorescence imaging and NIR-II bioluminescence imaging of peritoneal metastases (No. 1–7) Scale bar, 1 cm. **g** T/N ratios of corresponding serial number tumors in **f**. Black dashed line indicates the Rose criterion. Bars represent mean ± s.d. derived from *n* = 3 independent measurements. **h** H&E staining results of metastatic tumors (No. 1–7) margin in **f**. All borderlines between early metastatic lesions and normal tissues can be observed. All the metastases were confirmed to be malignant and repeated for three times in independent experiments. Scale bars, 0.2 mm. Source data underlying **b**, **e** and **g** are provided as a Source Data file.

signals were observed in the tumor site (Fig. 7a), with a high T/N ratio at 35 (Fig. 7b). The result was seven-fold higher than the standard Rose criterion (SNR = 5) and offered a proper manner to differentiate tumors from healthy tissues[50–52]. We compared the result with that of conventional bioluminescence imaging (termed as VIS bioluminescence) and NIR-II fluorescence imaging (Fig. 7a). As shown in Fig. 7b, the T/N ratios of these two imaging modes were 54.4% and 84.6% smaller than that of NIR-II bioluminescence imaging, respectively.

We finally investigated the potential application of NIR-II-BPs in metastatic tumor tracing. In a lymph node metastasis model (Fig. 7c), we implemented NIR-II fluorescence imaging and NIR-II bioluminescence imaging simultaneously. During fluorescence imaging, popliteal lymph node metastasis (POM) (Supplementary Figs. 17 and 18) and whole lymphatic vessels were observed (Fig. 7d). However, sacral lymph node metastasis (SCM) (Supplementary Figs. 17 and 18) was hardly identified due to the strong tissue autofluorescence (Fig. 7e). By contrast, POM and SCM were manifested clearly through NIR-II bioluminescence

imaging (Fig. 7d, e). Taking advantage of the sensitive ATP-mediated property, a 33-fold enhancement of T/N ratio (83.4) over NIR-II fluorescence imaging (2.5) also showed superior accuracy in popliteal lymph node metastases tracing (Fig. 7E). To validate the tumor tracing ability of NIR-II-BPs in more complicated physiological environments, we further tested them in mice bearing human ovarian adenocarcinoma peritoneal metastases (*n* = 3). 8 h later after administration of NIR-II-BPs, which enables the highest enrichment of NIR-II-BPs in the metastases, D-luciferin was injected into the mouse via tail vein (Supplementary Fig. 19). Through NIR-II fluorescence imaging, four tumors were barely discriminated with low T/N ratios at ~4. Besides, the borders between tumors and normal tissues were too fuzzy to tell apart (Fig. 7f). On the contrary, these metastases could be clearly visualized with high T/N ratios above 25 in NIR-II bioluminescence imaging (Fig. 7g). Moreover, three more tiny metastases (1–2 mm) were further identified with high precision (T/N > 17.3) (Fig. 7f, g). After removing all the tumors under the guidance of NIR-II bioluminescence imaging (Supplementary

Fig. 20), H&E staining was performed and confirmed the precise delineation of the tumor margin by the sharp cutting edge (Fig. 7h).

## Discussion

Bioluminescence imaging in the state-of-the-art NIR-II region with high penetration depth and high resolution is promising for in vivo biomedical imaging and has long remained a challenge. This work illustrates the feasibility of engineering the BRET and FRET processes to develop biocompatible NIR-II biolumines-cence probes (NIR-II-BPs). To achieve this, a well-designed heptamethine cyanine dye FD-1029 was synthesized. Since FD-1029 can overcome the concentration quenching problem that usually exists in conventional cyanine dyes, the as-prepared NIR-II-BPs can successfully extend bioluminescence from VIS region to NIR-II region, which benefit from the reduced scattering and absorption coefficient for deep tissue imaging. Moreover, this self-illuminating modality completely eliminates the adverse effects from external excitation light during in vivo imaging. This allows us to realize high-resolution imaging of vessels and lymphatics as well as tumors and metastases. Additionally, by carefully choosing the fluorophores, the BRET-FRET method can be a universal approach to realize bioluminescence with diverse emission wavelengths, as a distinct perspective to fill a growing demand for high contrast multicolor imaging and sensing.

Along with the undoubted advantages, bioluminescence imaging also presents some drawbacks, such as low emission efficiency, the requirement for unique substrates, environment-sensitive enzymes and limited emitting wavelengths traditionally in VIS and NIR-I region[53]. Although our probes presented here successfully address the issue of limited emitting wavelength and extend the conventional bioluminescence into the NIR-II region, other issues still remain to be resolved. These shortcomings may cause the requirement of long imaging time, multiple injections of the substrate, and limited biological environments during the in vivo NIR-II imaging. Notably, the bioluminescence imaging demonstrated here is based on passive targeting, which inevitably limits its use in more complex situations. Therefore, our future work will focus on the development of transfectable NIR-II bio-luminescence probes with enhanced flexiblity and efficiency. Nevertheless, the NIR-II-BPs still showed a bright prospect in real-time, deep tissue imaging with high contrast.

## Methods

**Materials**. All solvents including methanol, ethanol, petroleum ether, CH₂Cl₂ were provided by Titan Scientific. 1,2-distearoyl-sn-glycero-3-phosphoethanolamine-*N*-[methoxy-(polyethylene glycol)-2000] (DSPE-PEG2000), 1,2-distearoyl-sn-glycero-3-phosphoethanolamine-*N*-[carboxy-(polyethylene glycol)-2000] (DSPE-PEG2000-COOH) were provided by Avanti Polar Lipids.

**Preparation of NIR-II-BPs**. The detailed preparation procedures and methods of Cy5, Cy7.5 and FD-1029 are available in the supplementary information (Supplementary Figs. 1 and 2). NIR-II-BPs were prepared in two steps. Firstly, we prepared dye co-loaded micelles and next we conjugated the luciferase to the micelles.

The micelles were prepared by using modified film hydration technique. In a typical procedure, organic dyes were firstly dissolved in methanol to obtain a stock solution with a concentration of 5 mM. Different amount of organic dye stock solution was mixed with 1.2 mL DSPE-PEG2000 (25 mg mL⁻¹ in CHCl₃) and 0.4 mL DSPE-PEG2000-COOH (25 mg mL⁻¹ in CHCl₃). The solvent was removed by vacuum rotary evaporation to form a dry dye-containing lipid film. The dried film was hydrated with 10 mL deionized water at 60 °C and sonication for 30 s to make the clear micelles solution with different dye concentrations. The micelle solution was further concentrated by using a 100 KDa Amicon Ultra filter (Millipore Corporation) under centrifugation at 2000×g for 5 min. Finally, micelles are redissolved in 4 mL 1X Tris Buffer.

The micelles obtained after the centrifugation was then conjugated with firefly luciferase. In a typical conjugation reaction, 0.01 mL of micelles solution (~36 nmol mL⁻¹) was mixed with 6 mL HEPES buffer and 50 μL of freshly prepared aqueous solution with EDC (10 mg) and NHS (10 mg) was added. Then the

solution was magnetically stirred for 1 h at room temperature. After that, the mixture was reacted with 0.1 mL of amine-containing luciferase at ~1 mg mL⁻¹ over night at room temperature. The uncoupled free luciferase and excess EDC and NHS were removed by four washes using a 100 KDa Amicon Ultra filter (Millipore Corporation) under centrifugation at 2000×g for 5 min. The as-prepared NIR-II-BPs were kept in 1 mL PBS buffer for stocking at 4 °C. NIR-II-BPs solution was also lyophilized and determined as 13.3 mg mL⁻¹.

**Characterizations**. Transmission electron microscopy (TEM) measurements were carried out on a JEM 2100F microscope (Japan) operated at 200 kV. The samples were collected by using copper grids covered with carbon films for measurements. Absorption spectra were collected by using a PerkinElmer Lambda 750S UV-visible-NIR spectrometer at a 2000-nm/min scan rate. Bioluminescence and fluorescence spectra were recorded on Edinburgh Fluorescence Spectrometer FLS980 instrument. Excitation source for fluorescence spectra and imaging was an external semiconductor laser (Changchun New Industries Optoelectronics Tech. Co., Ltd.). Spectral data were further analyzed with Originlab 2017 and Microsoft Excel 2019. A home-built imaging system with 640 × 512-pixel 2D InGaAs NIRvana-LN camera was used in the bioluminescence and fluorescence imaging. All images were background and blemish corrected within the LightField imaging software, and analyzed and processed with Matlab. Unless otherwise specified, all spectra and imaging pictures were collected under identical experimental conditions. NMR spectra were measured with a Bruker AV 400 NMR spectrometer. The high-resolution mass spectra (HRMS) were measured within ESI mode using Bruker maxis UHR-TOF mass spectrometer. The MALDI-TOF-MS spectra were acquired on 5800 AB SCIEX. Thermal images were collected by using FLIR A315. Images and data were processed with the Mentor ResearchIR 3.0 software.

**Tissue phantom imaging study**. In all, 1% Intralipid was chosen as a simulated tissue due to its similar scattering characteristics. Glass capillary tube filled with 10 μg mL⁻¹ NIR-II-BPs was encapsulated for imaging. The capillary tube was then placed under a cylindrical culture dish and covered with different volumes of 1% Intralipid. 650 nm long-pass (LP), 850 nm (LP) with 1000 nm short-pass (SP), and 1000 nm LP filters were used.

**Photothermal effects comparison**. We chose three light sources (808 nm, 980 nm and 1064 nm), which are able to excite NIR-II-BPs, to study its photothermal effects. The power and duration of the laser excitations used in photothermal experiment are based on the latest guidelines produced by International Com-mission on Non-Ionizing Radiation Protection (ICNIRP)[43]. The guidelines indicate that the laser radiation exposure limits for the skin from wavelength 700–1050 nm obey a formulation $0.2 \times 10^{(0.002(\lambda-700))}$ W cm⁻² with an exposure duration from 10 s to 30000 s. Hence, 808 nm (329 mW cm⁻²), 980 nm (726 mW cm⁻²) and 1064 nm (1070 mW cm⁻²) are used in both in vitro and in vivo study (Fig. 3).

**Study of inhomogeneous illumination**. Distribution of the excitation light may bring distorted signals, especially in wide-field fluorescence imaging. A normal laser beam is usually subjected to a Gaussian point spread function (PSF). The rather low-quality factor (QF < 40) indicates a quite intensity variation of the excitation light[45], which would inevitably cause inhomogeneous signals. To study how inhomogeneous illumination affect the imaging results, 10 μg mL⁻¹ NIR-II-BPs in the capillary is used in in vitro study (Fig. 4a) and 200 μL 10 μg mL⁻¹ NIR-II-BPs was intradermal injected to mouse hind paw in in vivo study (Fig. 4f).

**Calculation of illumination light quality factor**. A three-step quality factor cal-culation (QF) was adopted according to the previous work[45]. In the first step, illumination responses were modeled by segmenting each intensity profile (two diagonal and one horizontal) into three linear functions. This method captured consistent illumination in each scan by estimating three lines with similar inter-cepts and near-0 slopes. A Gaussian illumination response generally yielded an initial line with a positive slope, a terminal line with a negative slope, and an intermediate line with near-0 slope. A perfect field illumination would yield three flat lines with 0 slopes. In the second step, root mean squared intensity (RMSI) and the length-adjusted absolute values of each of the three slopes were totaled across measurements, resulting in four variables for each light source tested. These variables were augmented by two more: total absolute first differences (AFDs) and total absolute second differences (ASD). The non-RMSI variables were then log transformed to make their distributions more Gaussian. The mean intensity, the aforementioned six variables, and their squares were calculated and further introduced into the regressions against the expertly ranked training set order. A scoring function came out representing the evenness of illumination. The third step was to turn the scoring function into a more familiar quality-rating score or QF as described in Eq. (1).

$$QF = 100 \times \min(1, \min(1, (\text{scoring function} - 24)/71)) \qquad (1)$$

**Cell cytotoxicity assay and cell imaging**. Human ovarian carcinoma cells (CAOV-3) were provided by American Type Culture Collection (human

embryonic kidney cell HEK293T). Mycoplasma contamination was excluded by mycoplasma removal kit. CAOV-3 were cultured in Dulbecco's Modified Eagle medium (DMEM) supplemented with 10% FBS and 1% Penicillin-Streptomycin at 37 °C in a humidified atmosphere of 5% $CO_2$. CAOV-3 were cultured in a 96-well plate ($8 \times 10^3$ cells/well) after 12 h incubation, the medium was replaced with 100 μL of fresh DMEM containing NIR-II-BPs with concentrations of 5, 10, 20, 40, 60, 80, 100, and 200 μg mL$^{-1}$, respectively. Cells were incubated with NIR-II-BPs for 12 h. In order to detect the cytotoxicity, 10 μL of Cell Counting Kit-8 (CCK-8) solution was added to each well of the microliter plate and the plate was incubated in the $CO_2$ incubator for additional 2 h. Dehydrogenase activities in living cells induce the formation of an orange-color formazan dye. The quality was assessed colorimetrically by using a multi-reader (TECAN, Infinite M200, Germany). The measurements were based on the absorbance values at 450 nm. Following formula was used to calculate the viability of cell growth:

Viability (%) = (mean absorbance value of treatment group/mean absorbance value of control group) × 100%.

For cell imaging experiments, CAOV-3 and human embryonic kidney cells (HEK293T, human embryonic kidney cell) ($1 \times 10^5$ cells per dish) were seeded in coverglass bottom dishes (35 mm × 10 mm), and then treated with the NIR-II-BPs at concentration of 10 μg mL$^{-1}$. After incubation for 3 h, the media were changed. 0.5 mL of DAPI (4, 6-diamidino-2-phenylindole) solution in PBS was added, and incubated for 15 min to stain the nuclei. After the incubation, the cells were softly washed twice to remove excessive DAPI. At last, 1 mL of PBS solution was added and the cells were visualized under a confocal laser scanning microscope (FluoView FV1000, Olympus). The fluorescence images were taken under 60× oil-immersion objective, the emission wavelength was set as 550(30) nm and over 800 nm for taking the bioluminescence of luciferase and NIR-II-BPs.

**Animal handling**. All animal experiments were performed under the approval of Fudan University's Administrative Panel on Laboratory Animal Care. Five-week-old female nude mice were purchased from Shanghai SLAC Laboratory animal CO. Ltd. Before imaging, all mice were anesthetized through the abdominal injection of urethane.

**Tumor model**. Subcutaneous CT-26 tumor model was prepared as follows. CT-26 tumor cells were harvested by centrifugation and resuspended in sterile PBS. CT-26 cells ($5 \times 10^7$ cells mouse$^{-1}$) were implanted subcutaneously into the right foreleg of BALB/c nude mice. When the tumors reached 1–2 cm in diameter (21–28 days after implant), the tumor-bearing mice were subjected to dual-channel imaging study.

Xenograft CAOV-3 tumor model was prepared as follows. CAOV-3 tumor tissues about 0.2 cm in diameter were implanted subcutaneously into the right fore-leg of BALB/c nude mice. When the tumors reached 0.5–0.7 cm in diameter (14–21 days after implant), the tumor-bearing mice were subjected to imaging studies.

A lymph node metastasis model was prepared as follows. CAOV-3 tumor cells were harvested by centrifugation and resuspended in sterile PBS. Then CAOV-3 cells ($5 \times 10^7$ cells mouse$^{-1}$) were intradermally injected to both hind paws of 5-week-old BALB/c nude mice. Popliteal lymph node and sacral lymph node metastasis were observed in sacrificed mice three weeks later.

Peritoneal ovarian metastasis model was prepared as follows. CAOV-3 tumor cells were harvested by centrifugation and resuspended in sterile PBS. Then CAOV-3 cells ($5 \times 10^7$ cells mouse$^{-1}$) were intraperitoneally injected into five-week-old BALB/c nude mice. Peritoneal ovarian metastasis was observed in sacrificed mice three weeks later.

**Imaging system**. The imaging system is illustrated in Supplementary Fig. 9A. All NIR images were collected with a 640 × 512 pixel 2D InGaAs CCD camera (Princeton Instruments, NIRvana-LN). During imaging, the InGaAs camera was cooled to −190 °C with liquid nitrogen, analog to digital conversion speed was set to 250 kHz and the exposure time was set to 10000 ms and 100 ms for NIR-II bioluminescence imaging and NIR-II fluorescence imaging respectively. An 850 nm LP filter (Edmund optics), a 1000 nm SP filter (Edmund optics) and a 1000 nm LP filter (Edmund optics) was used to choose the proper wavelength of NIR-II light. In addition, unless further described, NIR-II fluorescence imaging for comparison was carried out with an 808 nm 100 mW cm$^{-2}$ laser. VIS bioluminescence imaging was captured with a VIS imaging system (Berthold Technologies). During imaging, the camera was cooled to 10 °C and the exposure time was set to 100 ms.

**In vivo blood vessels and lymphatics imaging**. Mice were firstly anaesthetized and placed on a heating plate and injected with 200 μL of mixture of 10 μg mL$^{-1}$ NIR-II-BPs, 10 μM ATP and 10 μM D-luciferase. The blood vessels and lymphatics imaging was performed immediately after the injection of NIR-II-BPs. All groups within the study contained $n = 3$ mice. In order to compare imaging performance accurately, the signals of bioluminescence imaging and fluorescence imaging were normalized based on the maximum intensity of each picture.

**ATP-mediated NIR-II-BPs for tumor imaging and tumor tracing**. The bioluminescent ability of NIR-II-BPs towards different concentrations of ATP was studied before the tumor imaging (Supplementary Fig. 11).

The subcutaneous CAOV-3 tumor-bearing mice ($n = 3$) were first intravenously injected with NIR-II-BPs (200 μL, 100 μg mL$^{-1}$). In order to accumulate the NIR-II-BPs in tumor site, a second injection of D-luciferin (200 μL, 10 μM) was applied after different time periods. The best time for the second injection was confirmed to be 24 h after the first injection (Supplementary Fig. 15). Immediately after that, xenografted tumors can be observed under InGaAs camera (Supplementary Fig. 9A) through NIR-II bioluminescence imaging. In addition, VIS bioluminescence imaging was carried out similarly and obtained with a VIS imaging system (Supplementary Fig. 9A).

Lymph node metastasis bearing mouse was intradermally injected with a mixture of NIR-II-BPs (50 μL, 100 μg mL$^{-1}$) and D-luciferin (50 μL, 10 μM). NIR-II fluorescence imaging and NIR-II bioluminescence imaging was carried out simultaneously within 10 min. During simultaneous imaging, the laser power used to excite fluorescence was adjusted (808 nm, ~1 mW cm$^{-2}$) so that signals from both fluorescence and bioluminescence imaging can be collected by the apparatus. The lymph node metastatic tumors were removed under InGaAs camera with the direction of NIR-II-BPs (Supplementary Fig. 17). All collected tissues were further analyzed by H&E staining (Supplementary Fig. 18).

Peritoneal ovarian metastasis bearing mice ($n = 3$) were first intravenous injected with NIR-II-BPs (200 μL, 100 μg mL$^{-1}$). In order to get the maximum accumulation of NIR-II-BPs in metastasis, a second injection of D-luciferin (50 μL, 10 μM) was performed after different time periods. The best time for the second injection was confirmed to be 8 h after the first injection (Supplementary Fig. 19). It is worth noting that this time period is much shorter than that in previous xenografted tumor model. This should be attributed to the discrepancy in nanoprobe transport capabilities caused by the difference in tumor growth stage (Fig. 7h and Supplementary Fig. 16) and implantation cite in these two models[54]. NIR-II fluorescence imaging and NIR-II bioluminescence imaging was carried out immediately. The metastatic tumors were removed under InGaAs camera with the direction of NIR-II-BPs (Supplementary Fig. 20). All collected tissues were further analyzed by H&E staining (Fig. 7h).

**Reporting summary**. Further information on research design is available in the Nature Research Reporting Summary linked to this article.

## Data availability
The source data underlying Figs. 1–4 are provided as a Source Data file. Data supporting the findings of this study are available within the article and the associated Supplementary Information. Any other data are available from the corresponding authors upon reasonable request.

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

## Acknowledgements

The work was supported by the National Key R&D Program of China (2017YFA0207303), National Natural Science Foundation of China (NSFC, 21725502, 51961145403), and Key Basic Research Program of Science and Technology Commission of Shanghai Municipality (17JC1400100, 19490710100).

## Author contributions

F.Z., Y.F., and L.L. designed the study. L.L., B.L., S.D., S.W., C.S., and M.Z. performed the synthesis of probes; L.L., B.L., and S.W. analyzed data; and L.L., S.D., Y.F., C.Z. and F.Z. wrote the paper.

## Competing interests

The authors declare no competing interests.
