## [Peer Review File · Nature Communications]

REVIEWER COMMENTS

Reviewer #1 (Expertise: design of chemical imaging probes, Remarks to the Author):

This manuscript describes a self-illuminating NIR-II probe by using BRET-FRET-FRET cascade process. The authors designed an NIR-II fluorophore (FD1029) based on heptamethine cyanine with hydrophilic sulfonate groups. Then they prepared luciferase-modified micelles containing with Cy5, Cy7.5, and FD1029 as a self-illuminating NIR-II probe. By adding luciferase substrates, luminescence signals were observed at NIR-II regions beyond 1000 nm. The comparative experiment with fluorescence imaging revealed that NIR-II bioluminescence imaging has superior signal-to-noise ratio because of the reduced background signals. The authors also showed *in vivo* imaging of blood vessels, lymphatics, and tumor metastases with higher imaging contrast.

This study first demonstrates the utility of self-illuminating NIR-II probe for *in vivo* imaging. Owing to the absence of external excitation light, the images were obtained with high signal-to noise ratio. The enhanced contrast enables high-resolution imaging of blood vessels, lymphatics, and tumors. Although there are still challenges in current bioluminescence imaging system, this work would be acceptable for publication in Nature Communications after addressing the following concerns.

The authors rationally explained the reason of less self-aggregation of FD1029 by the large dihedral angle in ipsilateral indole groups and the increased hydrophilicity. The experimental results in Figure 2B-C and Supplementary Figures 1A-D showed the concentration dependency of the absorption and fluorescence signals. However, the local concentrations of the dyes in the micelle of NIR-II-BP were significantly high (9147 nmol/mL in Supplementary Table 2). I am concerned that the quenching effect still occurs in such a high local concentration. I suggest that the authors can measure the fluorescence spectra of FD1029 at higher concentration ranges, which would show the aggregation effect in a micelle.

Related to previous comment, the authors should describe how to determine the dye concentrations in a micelle (Supplementary Table 2) from spectroscopic data.

In tumor imaging experiments in Figures 7A and F, the timing of D-luciferin injection should be optimized. The bioluminescence images of xenograft tumor (Supplementary Figure 13) show 24 h is the best, while the images of peritoneal metastases (Supplementary Figure 16) show 8 h is the best. The requirement of the repetitive injection would be one limitation in the current system. The authors should mention the cause of the differences (the probe delivery rate by EPR, the probe stability in the local environment *in vivo*, or other effects).

Reviewer #2 (Expertise: Cancer imaging, probe development, Remarks to the Author):

This manuscript authored by Lu L et al. reports novel NIR-II bioluminescence molecular probes for in vivo imaging. NIR-II imaging is a highly promising technology. It is great to see that the authors further expand this field and achieved NIR-II bioluminescence imaging. The study is highly novel and interesting. The whole study includes the design of the probes, chemical synthesis, and in vitro and in vivo evaluation. The conclusion is well supported by the data presented. The manuscript can be accepted after addressing the following minor concerns.

- 1). Luciferases are on the surface of the nanoparticle. While Cy5, Cy7.5, and FD-1029 are encapsulated insides of the nanoparticles. Short space distance between donor-acceptor molecules is usually required for FRET and BRET. It is interesting to see the probes can still produce good NIR-II signals even using several bridge molecules. It would be helpful to discuss this.
- 2). How's the stability of the nanoprobes prepared in mouse serum or blood, besides in PBS?
- 3). It would be helpful to discuss the potential limitations of the probes designed.
- 4). NIR-II imaging has been evaluated in clinical studies and demonstrates advantages over traditional NIR-I imaging technique. It would be helpful to cite the relevant work (Nature Biomedical Engineering, 2020) to highlight the potential of NIR-II imaging.
- 5). The quality of some Figures can be further improved.
- 6). For animal imaging results shown in Figures, add the number of mice used in the studies.

Point-to-point response to reviewers

Reviewer 1:

This manuscript describes a self-illuminating NIR-II probe by using BRET-FRET-FRET cascade process. The authors designed an NIR-II fluorophore (FD1029) based on heptamethine cyanine with hydrophilic sulfonate groups. Then they prepared luciferase-modified micelles containing with Cy5, Cy7.5, and FD1029 as a self-illuminating NIR-II probe. By adding luciferase substrates, luminescence signals were observed at NIR-II regions beyond 1000 nm. The comparative experiment with fluorescence imaging revealed that NIR-II bioluminescence imaging has superior signal-to-noise ratio because of the reduced background signals. The authors also showed in vivo imaging of blood vessels, lymphatics, and tumor metastases with higher imaging contrast.

This study first demonstrates the utility of self-illuminating NIR-II probe for in vivo imaging. Owing to the absence of external excitation light, the images were obtained with high signal-to-noise ratio. The enhanced contrast enables high-resolution imaging of blood vessels, lymphatics, and tumors. Although there are still challenges in current bioluminescence imaging system, this work would be acceptable for publication in Nature Communications after addressing the following concerns.

Comments 1. *The authors rationally explained the reason of less self-aggregation of FD1029 by the large dihedral angle in ipsilateral indole groups and the increased hydrophilicity. The experimental results in Figure 2B-C and Supplementary Figures 1A-D showed the concentration dependency of the absorption and fluorescence signals. However, the local concentrations of the dyes in the micelle of NIR-II-BP were significantly high (9147 nmol/mL in Supplementary Table 2). I am concerned that the quenching effect still occurs in such a high local concentration. I suggest that the authors can measure the fluorescence spectra of FD1029 at higher concentration ranges, which would show the aggregation effect in a micelle.*

Response:

Thanks for the reviewer's valuable comments. For the high local concentrations of the dyes in the micelle, after careful double-check and recalculation, we found that all the calculations were correct (the method to determine the concentrations of the dyes and the raw data were shown in the answer to comment 2) except that an error occurred in the final unit conversion process, which caused the dye concentrations shown in Supplementary Table 2 to be three orders of magnitude larger than the actual values.

Specifically, when calculating the concentration of luciferase protein and DSPE-PEG2000, the mass concentration ($\mu\text{g mL}^{-1}$) and the molar concentration in (nmol mL^{-1}) were used respectively. However, the concentrations of the three dyes were calculated with standard samples in (μM). The error occurred in the final data consolidation process, which we took the unit (nmol mL^{-1}) as (nM) by mistake and thus expanded the dye concentrations by three orders of magnitude (nmol mL^{-1} is actually the same unit as μM , not nM). We apologize for the trouble and inconvenience caused by this error, and we have corrected this error with the actual values in the revised manuscript (Supplementary Table 2).

NIR-II-BPs compositions	Concentration (nmol·mL ⁻¹)
Micelles ^[a]	0.36 ± 0.02
DSPE-PEG2000 ^[b]	32.4 ± 1.8
Cy5 ^[c]	0.95 ± 0.01
Cy7.5 ^[c]	2.14 ± 0.10
FD-1029 ^[c]	9.15 ± 0.22
Luciferase ^[d]	1.76 ± 0.12

Supplementary Table 2. The concentration of NIR-II-BPs compositions. [a] The micelle concentration is estimated by the concentration of DSPE-PEG2000 taking account an aggregation number near 90.12. [b] The concentration of DSPE-PEG2000 is determined by colorimetric method (Supplementary Fig. 3A). [c] The concentration of organic dyes are determined by spectrophotometric method. [d] The concentration of luciferase is determined by Bradford assay (Supplementary Fig. 3B). The ratio of Luciferase to the micelles was ~5.

For the concentration quenching problem that the reviewer is concerned about, this situation indeed occurs in micelles loaded with high concentrations of dyes. For example, as shown in Supplementary Figures 1A-B, FD-1029 in micelles leads to significant concentration quenching when its concentration is above 10 μM . However, after the correction of the above error in Supplementary Table 2, the actual concentration of FD-1029 in our NIR-II-BPs was about 9 μM , which falls within the concentration range of our control experiment in Figure 2B-C and Supplementary Figures 1A-B. Therefore, we are sure that there is no significant aggregation quenching in our NIR-II-BPs. To make it clear we revised the related description in our manuscript. (Page 4, paragraph 1, line 9 to line 16) : "By increasing the dye concentration loaded in micelles, we found that the absorption and emission peaks of FD-1029 stayed constant in a large concentration range (1-10 μM), which was mainly attributed to the large dihedral angle ($\sim 60^\circ$) between ipsilateral indole groups in FD-1029 (Supplementary Fig. 1) and its good hydrophilicity. However, aggregation happens when higher concentration of FD-1029 is loaded ($> 10 \mu\text{M}$) (Fig. 2B and 2C). Therefore, the optimal concentration of the dyes is critical for modulating the superior property of the final NIR-II-BPs."

Supplementary Figure 1. (A) Absorbance spectra of FD-1029 in DSPE-PEG2000 micelles with the concentration of 1, 2, 5, 10 μM (from bottom to top). The dashed line refers to 12 μM FD-1029 in micelles. (B) Fluorescent emission spectra of FD-1029 in DSPE-PEG2000 micelles with the concentration of 1, 2, 5, 10 μM (from bottom to top). The dashed line refers to 12 μM FD-1029 in micelles.

Comments 2. Related to previous comment, the authors should describe how to determine the dye concentrations in a micelle (Supplementary Table 2) from spectroscopic data.

Response:

Thanks for your comment. The dye concentrations in a micelle are determined through a traditional spectrophotometric method based on Lambert-Beer law. Specifically, the absorption spectra of a series of standard solutions with different concentrations of dyes was first measured. Then the intensities of the corresponding characteristic absorption peaks of the dyes in NIR-II-BPs were measured. Finally, the concentration of each dye in the probe was determined in the plotted standard Absorbance-Concentration curve chart. We adopted the reviewer's opinion and added the related spectroscopic data and detailed data description in the revised manuscript (Supporting Information Figure S3).

Supplementary Figure 3. (A) Determination of the DSPE-PEG2000 in NIR-II-BPs with a colorimetric method. As the molar mass of DSPE-PEG2000 is $2805.5 \text{ g mol}^{-1}$, molar concentration of DSPE-PEG2000 in NIR-II-BPs is $32.4 \pm 1.8 \text{ nmol mL}^{-1}$. (B) Determination of the concentration of luciferase protein in NIR-II-BPs using the Bradford assay. As the molar mass of luciferase is $\sim 62 \text{ KDa}$, molar concentration of luciferase in NIR-II-BPs is $1.76 \pm 0.12 \text{ nmol mL}^{-1}$. (C) The concentration of Cy5 in NIR-II-BPs was determined to be $0.95 \pm 0.01 \mu\text{M}$ with spectrophotometric method. (D) The concentration of Cy7.5 in NIR-II-BPs was determined to be $2.14 \pm 0.10 \mu\text{M}$ with spectrophotometric method. (E) The concentration of FD-1029 in NIR-II-BPs was determined to be $9.15 \pm 0.22 \mu\text{M}$ with spectrophotometric method.

Comments 3. In tumor imaging experiments in Figures 7A and F, the timing of D-luciferin injection should be optimized. The bioluminescence images of xenograft tumor (Supplementary Figure 13) show 24 h is the best, while the images of peritoneal metastases (Supplementary Figure 16) show 8 h is the best. The requirement of the repetitive injection would be one limitation in the current system. The authors should mention the cause of the differences (the probe delivery rate by EPR, the probe stability in the local environment in vivo, or other effects).

Response:

We would like to thank the reviewer for the comment. The reviewers have unique insights to discover this interesting phenomenon that NIR-II-BPs exhibited different enrichment processes in xenograft tumors and peritoneal metastases. In this work, we have optimized the injection time of D-luciferin by repeating the bioluminescence experiments of xenograft tumor and peritoneal metastases on three mice each (revised Supplementary Figs. 13 and 17) and the results were similar (8 h is the best for peritoneal metastases and 24 h is the best for xenograft tumor). We attribute these results to the different nanoprobe transport capabilities in different stages of tumor growth and different sites of implantation.

As we all know, the delivery of nanomaterials to tumors is a complex process that involves multiple variables. This process has received extensive attention and great research enthusiasm in the past three decades. Recent research has found that, in addition to the passive delivery process, EPR effect, discovered in the past, the active delivery process might play a dominant role in this process (Maeda H, Wu J, Sawa T, et al. *Journal of controlled release*, 2000, 65(1-2): 271-284; Sindhwani S, Syed A M, Ngai J, et al. *Nature Materials*, 2020: 1-10.). However, it is still commonly recognized that the complex microenvironment of the tumor site, many factors including tumor type and size, stage of disease (tumor growth rate), site of implantation, growth factor expression, and the host species etc., may affect the delivery efficiency and residence time of nanomaterials. (Schroeder A, Heller D A, Winslow M M, et al. *Nature Reviews Cancer*, 2012, 12(1): 39-50; Torchilin V. *Advanced drug delivery reviews*, 2011, 63(3): 131-135.).

In our study, two tumor models were used in imaging experiments. These two tumor models differ in the stages of tumor growth and the sites of implantation. Specifically, while xenograft tumors were constructed through subcutaneous tumor tissue implantations, peritoneal metastases were constructed through intraperitoneal injection of cancer cells. Firstly, these two models of tumor are in different stages of growth, which can be demonstrated not only by our experimental processes (tumor tissue implantation vs cell injection) but also by H&E stain results of metastatic tumors (Figure 7H) and xenograft tumors (revised manuscript Supplementary Figure 14). As shown in Figure 7H and Supplementary Figure 14, cell morphology in xenograft tumors is more disordered than in metastatic tumors, which means nanomaterials will have more chances to retain in the cell interstitium and stored for longer time in xenograft tumors (Maeda H, Matsumura Y. *Critical reviews in therapeutic drug carrier systems*, 1989, 6(3): 193-210; Nichols J W, Bae Y H. *Journal of Controlled Release*, 2014, 190: 451-464.). Secondly, the implantation site of tumors (subcutaneous vs peritoneal) can also influence the proliferation rates, vascular pore size, and tumor microenvironment (Lauk S, Zietman A, Skates S, et al. *Cancer research*, 1989, 49(16): 4557-4561; Wilhelm S, Tavares A J, Dai Q, et al. *Nature reviews materials*, 2016, 1(5): 1-12.). It was also found that tumor tissues

have a reduced vascular permeability in subcutaneous microenvironment as compared with the underlying tissues, which supported that the enrichment process of nanomaterials in xenograft tumors will be slower than that in metastatic tumors (Hobbs S K, Monsky W L, Yuan F, et al. *Proceedings of the National Academy of Sciences*, 1998, 95(8): 4607-4612.).

Combining the above two points, we believe that due to their earlier stage and implantation site in the abdomen, peritoneal metastases have better vascular permeability, higher lymphatic drainage capacity and much ordered cell morphology than subcutaneous xenograft tumors. These factors have led to more rapid accumulation and clearance of nanomaterials in peritoneal metastases, therefore shorter time period (8 h versus 24 h), than that in xenograft tumors,. This is indeed a very interesting phenomenon and worthy of further study.

Moreover, we adopt the suggestion of reviewers and make corresponding explanations in the revised manuscript so that readers will not be confused about this difference. (Page 23, paragraph 3, line 4 to line 8): "The best time for the second injection was confirmed to be 8 hours after the first injection (Supplementary Fig. 17). It is worth noting that this time period is much shorter than that in previous xenografted tumor model. This should be attributed to the discrepancy in nanoprobe transport capabilities caused by the difference in tumor growth stage (Fig. 7H and Supplementary Figure S14) and implantation cite in these two models."

Supplementary Figure 13. NIR-II bioluminescence imaging results of xenografted tumor bearing mice (n=3) after tail injection with NIR-II-BPs for various hours. Scale bar, 1 cm.

Supplementary Figure 17. NIR-II bioluminescence imaging results of peritoneal metastases bearing mice (n=3) after tail injection with NIR-II-BPs for various hours. Scale bar, 1 cm.

Fig. 7 (H) H&E staining results of metastatic tumors (No. 1-7) margin in F. All borderlines between early metastatic lesions and normal tissues can be observed. All the metastases were confirmed to be malignant. Scale bars, 0.2 mm.

Supplementary Figure 14. H&E staining results of xenografted tumor in Supplementary Fig. 13. The tumor tissue was confirmed to be malignant. Scale bars, 0.1 mm.

The repetitive injection of the substrate is indeed a limitation of NIR-II-BPs. To show more rigorous, we discussed this limitation in the discussion part at the end of the revised manuscript (page 14, paragraph 2, line 1 to line 10): "Along with the undoubted advantages, bioluminescence imaging also presents some drawbacks, such as low emission efficiency, the requirement for unique substrates, environment-sensitive enzymes and limited emitting wavelengths traditionally in VIS and NIR-I region.⁵² Although our probes presented here successfully address the issue of limited emitting wavelength and extend the conventional bioluminescence into the NIR-II region, other issues still remain to be resolved. These shortcomings may cause the requirement of long imaging time, multiple injections of the substrate, and limited biological environments during the in vivo NIR-II imaging. Notably the bioluminescence imaging demonstrated here is based on passive targeting, which inevitably limits its use in more complex situations. Therefore, our future work will focus on the development of transfectable NIR-II bioluminescence probes with enhanced flexibility and efficiency. Nevertheless, the NIR-II-BPs still showed a bright prospect in real-time, deep tissue imaging with high contrast."

Reviewer 2:

This manuscript authored by Lu L et al. reports novel NIR-II bioluminescence molecular probes for in vivo imaging. NIR-II imaging is a highly promising technology. It is great to see that the authors further expand this field and achieved NIR-II bioluminescence imaging. The study is highly novel and interesting. The whole study includes the design of the probes, chemical synthesis, and in vitro and in vivo evaluation. The conclusion is well supported by the data presented. The manuscript can be accepted after addressing the following minor concerns.

Comment 1. *Luciferases are on the surface of the nanoparticle. While Cy5, Cy7.5, and FD-1029 are encapsulated insides of the nanoparticles. Short space distance between donor-acceptor molecules is usually required for FRET and BRET. It is interesting to see the probes can still produce good NIR-II signals even using several bridge molecules. It would be helpful to discuss this.*

Response:

We thank the reviewer for the positive comments and suggestions. As the reviewer mentioned, the space distance between donor and acceptor is key to the energy transfer efficiency. Based on our experimental data, we calculated the BRET and FRET efficiencies of each step, and further calculated the Förster distances (R_0) of each FRET step (page 6 in the revised supplementary information). Generally, FRET occurs with a distance between the donor and the acceptor typically in the range of 1-10 nm. In our NIR-II-BPs, the R_0 between Cy5 & Cy7.5 and Cy7.5 & FD-1029 were calculated to be 1.3 nm and 6.4 nm respectively. Additionally, previous study also proved that the shell thickness of a DSPE-PEG2000 micelle was about 2 nm (Johnsson M, Hansson P, Edwards K. *The Journal of Physical Chemistry B*, 2001, 105(35): 8420-8430.), which further assured us the distance between Luciferase protein and Cy5 was also within 10 nm considering the random distribution of Cy5 in the micelle. Hence, we believe the distance between all donors and acceptors are short enough for efficient energy transfer.

In order to make the manuscript objective and rigorous, we take the reviewer's advice and discuss more details about the distance between donors and acceptors in the revised manuscript (page 5, paragraph 2, line 11 to line 14): "Under the circumstance, FRET efficiencies between Cy5&Cy7.5 and Cy7.5&FD-1029 were determined to be 64.1% and 90.8%. The Förster distances (R_0) of these two steps were also calculated to be 1.3 nm and 6.4 nm, respectively (Supplementary Figs. 4A and 4B), which guarantee the efficient FRET between these dyes and bridge the energy to the NIR-II region."

The detailed calculation procedure of the FRET efficiencies and the Förster distances were available in Supplementary Information (page 6): "Calculation of FRET efficiency and BRET ratio."

FRET efficiencies between organic dyes were calculated according to the equation:

$$E = 1 - F'D/FD,$$

where $F'D$ and FD are the donor fluorescence intensities with and without an acceptor, respectively.⁶ The optimal FRET efficiency between Cy5 and Cy7.5 was calculated to be 64.07%. The optimal FRET efficiency between Cy7.5 and FD-1029 was calculated to be

90.80%.

The Förster distances (R_0) of these two steps were also calculated to be 1.3 nm and 6.4 nm according to the equations:

$$J(\lambda) = \frac{[\int_0^{\infty} F_D(\lambda) \epsilon_A(\lambda) \lambda^4 d\lambda]}{[\int_0^{\infty} F_D(\lambda) d\lambda]}, R_0 = 0.0211[\kappa^2 n^{-4} \Phi_D J(\lambda)]^{1/6}$$

where $F_D(\lambda)$ is the area-normalized emission spectrum of donor, $\epsilon_A(\lambda)$ is the molar absorption spectrum of the acceptor in $M^{-1}cm^{-1}$, λ is the wavelength in nm, κ^2 is orientation factor ($\kappa^2=2/3$ due to dynamic averaging donor-acceptor systems), Φ_D is quantum yield of the donor, and $n=1.35$ is the refractive index of the surrounding medium.⁷

BRET ratio was defined by the acceptor emission relative to the donor emission.⁸ As illustrated in Supplementary Fig. 4F, area A is the integrated total emission (from 625 nm to 800 nm) from Cy5 and area B is the integrated total emission from luciferase (500-625 nm). Thus, the BRET ratio of luciferase-Cy5 is 4.3. Similarly, the BRET ratio of NIR-II-BPs was calculated to be 4.2 through dividing the intensity of acceptor emission (625-1400 nm) by that of the donor emission (500-625 nm).⁹

Comment 2. How's the stability of the nanoprobes prepared in mouse serum or blood, besides in PBS?

Response:

Thanks for the comment. The stability of the NIR-II-BPs is very important for *in vivo* imaging applications. Following this suggestion, we have further conducted the stability experiments of NIR-II-BPs in mouse serum and PBS buffer, including fluorescence stability, bioluminescence stability, and micelle size stability. The nanoprobes show good stability after storage in mouse serum for one week. These stability properties of NIR-II-BPs are added in the revised manuscript as following:

(Page 5, paragraph 3, line 1 to line 2): "Thanks to the lipid shells, the NIR-II-BPs were stable for a week in both PBS buffer and mouse serum without obvious aggregation (Supplementary Fig. 5)."

(Page 5, paragraph 3, line 5 to page 6, line 2): "Besides, NIR-II-BPs also exhibited good bioluminescence stability after stored in mouse serum for one week at various temperatures (Supplementary Figs. 6B-E)."

Related data can be found in Supplementary Figure 5 and Supplementary Figure 6.

Supplementary Figure 5. Dynamic light scattering (DLS) measurement of NIR-II-BPs in PBS buffer and mouse serum. (A) As-made NIR-II-BPs. (B) NIR-II-BPs stored in PBS buffer for one week. (C) NIR-II-BPs stored in mouse serum for one week.

Supplementary Figure 6. Stability of NIR-II-BPs. (A) NIR-II bioluminescence intensity of NIR-II-BPs (10 $\mu\text{g mL}^{-1}$) in PBS (pH = 7.4) over 24 hours. (B) NIR-II fluorescence intensity of NIR-II-BPs in mouse serum after storing at varied temperatures for 24 hours and for one week.

Error bars represent the standard deviation of three separate measurements. (C) NIR-II bioluminescence intensity of NIR-II-BPs in mouse serum after storing at varied temperatures for 24 hours and for one week. Error bars represent the standard deviation of three separate measurements.

Comment 3. *It would be helpful to discuss the potential limitations of the probes designed.*

Response:

Thanks for your comment. As the reviewer mentioned, NIR-II-BPs do have some drawbacks. They have some disadvantages existing in traditional bioluminescence probes, such as low emission efficiency, the requirement for unique substrates, and a steady biological environment. These shortcomings may cause the requirement of longer imaging time, multiple injections of the substrate, and limited biological environments during the *in vivo* NIR-II imaging. These problems will indeed limit the practical applications of the probes to a certain extent.

In order to make the manuscript objective and rigorous, we have discussed the drawbacks of bioluminescence in the discussion part at the end of the revised manuscript (page 14, paragraph 2, line 1 to line 10): "Along with the undoubted advantages, bioluminescence imaging also presents some drawbacks, such as low emission efficiency, the requirement for unique substrates, environment-sensitive enzymes and limited emitting wavelengths traditionally in VIS and NIR-I region.⁵² Although our probes presented here successfully address the issue of limited emitting wavelength and extend the conventional bioluminescence into the NIR-II region, other issues still remain to be resolved. These shortcomings may cause the requirement of long imaging time, multiple injections of the substrate, and limited biological environments during the *in vivo* NIR-II imaging. Notably, the bioluminescence imaging demonstrated here is based on passive targeting, which inevitably limits its use in more complex situations. Therefore, our future work will focus on the development of transfectable NIR-II bioluminescence probes with enhanced flexibility and efficiency. Nevertheless, the NIR-II-BPs still showed a bright prospect in real-time, deep tissue imaging with high contrast."

Comment 4. *NIR-II imaging has been evaluated in clinical studies and demonstrates advantages over traditional NIR-I imaging technique. It would be helpful to cite the relevant work (Nature Biomedical Engineering, 2020) to highlight the potential of NIR-II imaging.*

Response:

Thanks for the reviewer's suggestion. The advantages of NIR-II imaging over traditional NIR-I imaging techniques in clinical scenarios highlight the promising clinical potential of intraoperative NIR-II fluorescence imaging and NIR-II image-guided surgery. The relevant work (Nature Biomedical Engineering, 2020) really provides sufficient confidence for further researches to develop new NIR-II probes and apply them from bench to bedside. We cite this article in our revised manuscript as follows:

(Page 3, paragraph 1, line 6 to line 8): "Recent clinical study also highlights the promising clinical potential of intraoperative NIR-II fluorescence imaging and NIR-II image-guided surgery."

Comment 5. The quality of some Figures can be further improved.

Response:

Thanks for your comment and the instructive suggestion. We have improved the quality of the pictures in the revised manuscript, especially in Fig. 3 and Fig. 4. We have made improvements in both visibility and expressiveness of these figures, and hope that these improvements will help readers better understand the work.

Fig. 3 Comparing the autofluorescence and photothermal effect of external excitation in NIR-II bioluminescence and fluorescence imaging.

Fig. 4 Comparing the signal uniformity of NIR-II bioluminescence and fluorescence.

Comment 6. For animal imaging results shown in Figures, add the number of mice used in the studies.

Response:

Thanks for your comment. To show more rigorous, we have now added the description of mice number used during animal imaging experiments in the revised manuscript.

(Page 9, paragraph 2, line 7 to line 9): "Next, we performed real-time NIR-II bioluminescence imaging of the mice abdominal vascular and lymphatic (n = 3) after a mixture of NIR-II-BPs, ATP, and D-luciferin was applied via tail vein injection immediately."

(Page 11, paragraph 3, line 2): "Nude mice (n = 3) bearing subcutaneous human ovarian adenocarcinoma tumors were adopted as the model."

(Page 12, paragraph 2, line 11 to line 13): "To validate the tumor tracing ability of NIR-II-BPs in more complicated physiological environments, we further tested them in mice bearing human ovarian adenocarcinoma peritoneal metastases (n = 3)."

(Page 22, paragraph 4, line 1 to line 2): "The subcutaneous CAOV-3 tumor bearing mice (n=3) were first intravenously injected with NIR-II-BPs (200 μ L, 100 μ g mL⁻¹)."

(Page 23, paragraph 3, line 1 to line 2): "Peritoneal ovarian metastasis bearing mice (n=3) were first intravenous injected with NIR-II-BPs (200 μ L, 100 μ g mL⁻¹)."

REVIEWERS' COMMENTS:

Reviewer #1 (Remarks to the Author):

In the revised manuscript, the authors discussed concentration dependency of the dyes in micelles as the component of the bioluminescent probe and potential limitations of this bioluminescence probe. The authors also explained in detail about the differences of the best timing of the probe injection for each in vivo imaging experiments.

These modifications sufficiently addressed my concerns on the original manuscript. After addressing one minor comment below, I recommend the acceptance of this manuscript for publication in Nature Communications.

Minor comment

In Figure 2A, the chemical structure of the reactant N-[(3-(Anilinomethylene)-2-chloro-1-cyclohexen-1-yl)methylene]aniline monohydrochloride should be added in the scheme.

Reviewer #2 (Remarks to the Author):

The authors addressed the reviewer's comments and the revised manuscript is acceptable for publication.

Point-to-point response to Reviewers

Reviewer #1 (Remarks to the Author):

1. In the revised manuscript, the authors discussed concentration dependency of the dyes in micelles as the component of the bioluminescent probe and potential limitations of this bioluminescence probe. The authors also explained in detail about the differences of the best timing of the probe injection for each in vivo imaging experiments.

These modifications sufficiently addressed my concerns on the original manuscript. After addressing one minor comment below, I recommend the acceptance of this manuscript for publication in Nature Communications.

A: Thanks for your comment. Once again, we would like to appreciate the referees for reviewing our manuscript and putting forward many constructive and valuable comments.

2. In Figure 2A, the chemical structure of the reactant N-[(3-(Anilinomethylene)-2-chloro-1-cyclohexen-1-yl)methylene]aniline monohydrochloride should be added in the scheme.

A: Thanks for your useful suggestion. We have now added the chemical structure of reactant N-[(3-(Anilinomethylene)-2-chloro-1-cyclohexen-1-yl)methylene]aniline monohydrochloride in revised Fig. 2A.

Fig. 2A Synthetic route of FD-1029.

Reviewer #2 (Remarks to the Author):

1. The authors addressed the reviewer's comments and the revised manuscript is acceptable for publication.

A: Thanks for your comment. Once again, we would like to appreciate the referees for reviewing our manuscript and putting forward many constructive and valuable comments.